# New Construction of Functionalized CuO/Al_2_O_3_ Nanocomposite-Based Polymeric Sensor for Potentiometric Estimation of Naltrexone Hydrochloride in Commercial Formulations

**DOI:** 10.3390/polym13244459

**Published:** 2021-12-20

**Authors:** Amal M. Al-Mohaimeed, Gamal A. E. Mostafa, Maha F. El-Tohamy

**Affiliations:** 1Department of Chemistry, College of Science, King Saud University, P.O. Box 22452, Riyadh 11451, Saudi Arabia; muhemeed@ksu.edu.sa; 2Department of Pharmaceutical Chemistry, College of Pharmacy, King Saud University, P.O. Box 2457, Riyadh 11451, Saudi Arabia

**Keywords:** naltrexone hydrochloride, nanocomposites, polymeric sensors, commercial formulations

## Abstract

Electrically conductive polymeric nanocomposites with nanoparticles are adaptable types of nanomaterials that are prospective for various applications. The extraordinary features of copper oxide (CuO) and aluminium oxide (Al_2_O_3_) nanostructures, encourages extensive studies to prospect these metal oxide nanocomposites as potential electroactive materials in sensing and biosensing applications. This study suggested a new CuO/Al_2_O_3_ nanocomposite-based polymeric coated wire membrane sensor for estimating naltrexone hydrochloride (NTX) in commercial formulations. Naltrexone hydrochloride and sodium tetraphenylborate (Na-TPB) were incorporated in the presence of polymeric polyvinyl chloride (PVC) and solvent mediator *o*-nitrophenyloctyl ether (*o*-NPOE) to form naltrexone tetraphenylborate (NTX-TPB) as an electroactive material. The modified sensor using NTX-TPB-CuO/Al_2_O_3_ nanocomposite displayed high selectivity and sensitivity for the discrimination and quantification of NTX with a linearity range 1.0 × 10^−9^–1.0 × 10^−2^ mol L^−1^ and a regression equation E_mV_ = (58.25 ± 0.3) log [NTX] + 754.25. Contrarily, the unmodified coated wire sensor of NTX-TPB exhibited a Nernstian response at 1.0 × 10^−5^–1.0 × 10^−2^ mol L^−1^ and a regression equation E_mV_ = (52.1 ± 0.2) log [NTX] + 406.6. The suggested modified potentiometric system was validated with respect to various criteria using the methodology recommended guidelines.

## 1. Introduction

Advances in nanoscience technologies and nanomaterial engineering have opened up new areas in scientific research and the evolution of modified sensing and biosensing probes. The current studies have focused on the production of nanocomposites instead of single nanoparticles. These nanocomposites usually possess different nanoscale domains, which induce synergistic effects as a result of their interfacial interactions [1].

Nanocomposites hold both extraordinary characteristics of nanomaterials and polymer advantages including, chemical resistance, high conductivity, biocompatibility and elasticity [2]. The recent progress in scientific areas, such as chemical engineering [3], biochemistry [4] and physics [5], require the development of novel sensing technologies that combine lower power utilization and miniaturization with highly tactile sensitivity. Nanocomposites are also nanostructures with high activities that exhibit unusual significant combinations and unique engineering possibilities. With increasing the growth rate and rapid inquire to be in sensor constructions, their potential is so prominent that they are successfully utilized in various sensing and biosensing applications [6]. They have appeared as viable alternatives to control the drawbacks of micro composites. Furthermore, these materials are having extraordinary designs and unique optical characteristics that are not present in conventional composites [7]. Additionally, the nanocomposite synthesis is considered to be the main step in construction of various electronic devices [8], drug delivery systems [9], biomedical and immunosensing applications [10,11].

Currently, several reports are focused on metal oxides, including aluminium oxide (Al_2_O_3_), copper oxide (CuO), nickel oxide (NiO), etc. Moreover, more attention has been focused on the perspective of copper oxide in various aspects such as antibacterial agents [12], catalysis [13], food packaging [14], sensors [15] and medicine [16]. Recently, there has been a growing awareness of the development of aluminium oxide nanoparticles (Al_2_O_3_NPs) which have a high specific surface area with extraordinary optical, catalytic, thermodynamic stability properties for progressive engineering and industrial applications [17,18,19]. Few studies reported the use of CuO/Al_2_O_3_ in sensing applications such as Khan et al. [20] studied the electrochemical oxidation of ammonia over copper oxide impregnated γ-Al_2_O_3_ nanocatalysts. Also, the structural features, electrical characteristics and gas sensing applications of polymeric copper/alumina hybrid nanocomposite has been reported by Sankar et al. [21]. Another study conducted by using the p-Type copper aluminum oxide thin films for gas-sensing applications has been reported by Baratto et al. [22].

Some recent studies reported the synthesis of CuONPs and Al_2_O_3_NPs by various techniques, including pyrolysis, sol-gel, sputtering, laser ablation and electrochemical reduction [23,24,25,26,27,28,29,30,31,32]. Sodeifian and Behnood [33] reported a microwave-assisted method for the synthesis of CuO/Al_2_O_3_ nanocomposite for photocatalytic degradation of methylene blue removal from aqueous solution under UV irradiation.

Electrochemical techniques, including amperometry, conductometry and potentiometry are more reliable and economical techniques suitable for in-field of sensing, biosensing, chemical analysis and biomedical applications [34,35,36,37]. These techniques are also fast in terms of spending short analytical time, allowing on-line detecting of aqueous samples. The potentiometric technique is one of the most promising self-powered techniques in which the potential difference measurements are resulted from the accumulation of analytes under the electrostatic mechanism on the surface of the working electrode and the reference one [38]. Recently, various potentiometric sensors have been modified with metal, metal oxides and nanocomposites or coupled with biosensing electrodes to enhance their sensitivity and limits of detection [39,40]. Various dozens of compounds can be estimated with the potentiometric sensors that commonly contain membranes fabricated from high molecular weight polyvinyl chloride (PVC), plasticizers, including dioctyl sebacate (DOS), dioctyl phthalate (DOP), dibutyl sebacate (DBS), dibutyl phthalate (DBP) and *o*-nitrophenyl octyl ether (*o*-NPOE) as solvent mediator. Additionally, these membranes also formed from lipophilic ions or molecules, acting as active materials to produce specific analyte interaction in the membrane sites for pre-detection of the selectivity of the relating sensors [41].

The potentiometric coated wire sensors are commonly constructed from highly conductive metal wire such as silver, gold, platinum, copper and aluminium. The metal wire is used as a substrate which is coated with a polymeric cocktail containing the active sites of the selective membrane [42]. Sodium tetraphenylborate (Na-TPB) is a white powder that has the ability to catalyze the ion-exchange process via the interfaces of hydrophobic membrane when used with a suitable ion-pair complex [43].

Naltrexone hydrochloride (NTX), is an opioid medication antagonist, primarily used to control opioid and alcohol disorders (Figure 1). It has also been found to have potential in the case of euphoria associated with drug abuse disorders [44]. NTX was previously estimated and quantified by several analytical techniques, such as spectroscopic [45], chromatographic separation [46,47], and electrochemical methods [48]. Although these previously reported techniques exhibited acceptable sensitivity and selectivity for NTX detection, they still have certain limitations as they require long analytical separation time, high technical skills and consumption of large amounts of solvents.

The objective of this study was to design a modified metal oxide (CuO/Al_2_O_3_) nanocomposite coated wire sensor with ultra-sensitivity and selectivity towards the detection of NLX in its commercial products. A new strategy involving exploiting the unique physical, chemical, optical and conductive properties of the selected metal oxides has been suggested to enhance the sensitivity and selectivity of the potentiometric modified sensor. The incorporation of CuO/Al_2_O_3_ in polymeric matrix will play an impact role in the sensitivity and selectivity of the suggested sensor towards the determined drug. To ensure the analytical suitability of the suggested method, method validation is according to ICH guidelines [49]. Furthermore, a comparative study was carried out between the suggested CuO/Al_2_O_3_ nanocomposite coated membrane sensor and the conventionally designed type.

## 2. Materials and methods

### 2.1. Chemicals

Pure grade of opioid antagonist medication Naltrexone hydrochloride and Naltrexone hydrochloride^®^ tablets (50 mg/tablet) was gained from (Intas Pharmaceuticals Limited, Ahmedabad, India). Sigma Aldrich, Hamburg, Germany, supplied various analytical chemicals and solvents, including acetone 99.9%, methanol 99.9%, ethanol 99.9%, tetrahydrofuran (THF) 97.0%, *o*-NPOE, hydrochloric acid 37%, sodium tetraphenylborate (Na-TPB) and high molecular weight PVC. Copper chloride (99.0%), aluminium nitrate nonahydrate (99.9%) and sodium hydroxide (99.9%) were obtained from BDH (Poole, UK).

### 2.2. Instruments

This study was conducted using a “digital pH meter HANNA, model 211 (HANNA instruments, Rhode Island, Woonsocket, RI, United States) and the pH meter Metrohm pH-meter model 744 (Metrohm Co., Herisau, Switzerland) was used to adjust the pH of the sample solution throughout the experiments. The designed potentiometric system was consisted of a constructed conventional naltrexone hydrochloride-tetraphenyl borate (NTX-TPB) or modified NTX-TPB-CuO/Al_2_O_3_ nanocomposite coated wire sensor in conjunction with silver/silver chloride (Ag/AgCl) as reference one. The synthesized metal oxide nanoparticles and the nanocomposite were characterized using various spectroscopic and microscopic techniques, including UV-2450 spectrophotometer (Shimadzu Corporation, Kyoto, Japan), Fourier-Transform Infrared spectroscopy (FT-IR) Spectrum BX spectrometer (PerkinElmer, Waltham, Massachusetts, United States). X-ray diffraction (XRD) Shimadzu XRD-6000 diffractometer (Shimadzu, Kyoto, Japan), scanning electron microscope (SEM) JSM-7610F (JEOL Ltd., Tokyo, Japan), and a transmission electron microscope (TEM) JEM-2100F, (JEOL Ltd., Tokyo, Japan). Furthermore, Energy-Dispersive X-Ray Spectroscopy (EDX), using EDX-8100 (Shimadzu, Kyoto, Japan) analysis was applied to detect the presence of Cu, Al and O elements in the synthesized nanomaterials.

### 2.3. Preparation of NTX-TPB Electroactive Complex

The electroactive compound NTX-TPB was prepared by mixing equal volume (50 mL) of an equimolar concentration (1.0 × 10^−2^ mol L^−1^) of aqueous NTX and TPB solution. A white precipitate of NTX-TPB was formed. The mixture solution was filtered using Whatman filter paper No. 41 and the precipitate was washed using deionized water and left to dry overnight.

### 2.4. Synthesis of CuO and Al_2_O_3_ Nanoparticles

The precipitation method was used to synthesize CuO nanoparticles by dissolving 1.596 g of copper nitrate (Cu(NO_3_)_2_·3H_2_O) in 100 mL deionized water to form 0.1 mol L^−1^ concentration. An aqueous sodium hydroxide solution (0.1 mol L^−1^) was prepared and slowly added dropwise under vigorous stirring until the pH achieved 14. The formed precipitate was repeatedly washed three times and neutralized by deionized water and absolute ethanol. Subsequently, the resulting precipitate was dried at 80 °C for 12 h, then it was calcined for 4 h at 500 °C.

Al_2_O_3_ nanoparticles were prepared using a sol-gel method by heating a solution of citric acid/aluminium nitrate nonahydrate prepared in deionized water with a molar ratio of 0.5. The solution was heated at 60 °C under continuous stirring until the formation of white sol. The resulted gel was heated under constant stirring up to 80 °C until the formation of transparent gel. The formed gel was dried in an oven for 12 h at 90 °C and then ground and sintered at 600 °C for 4 h.

### 2.5. Synthesis of Polymeric NTX-TPB-CuO/Al_2_O_3_ Nanocomposite

The polymeric solution of CuO/Al_2_O_3_ nanocomposite was prepared by dissolving 5 mg of each previously prepared CuO and Al_2_O_3_ nanoparticles, 10 mg NTX-TPB ion pair, 190 mg of PVC and 0.35 mL *o*-nitrophenyloctyl ether in 5 mL THF under continuous stirring to form a polymeric cocktail of NTX-TPB-CuO/Al_2_O_3_ nanocomposite. Then, it was used to modify the surface of the suggested modified NTX-TPB-CuO/Al_2_O_3_ nanocomposite sensor.

### 2.6. Characterization of Synthesized Nanomaterials

Spectroscopic analysis for the synthesized CuO, Al_2_O_3_ and CuO/Al_2_O_3_ nanomaterials was performed at a wavelength range of 200–500 nm to confirm their formation. FT-IR detection was also conducted to determine the possible functional groups that can appear in the CuO, Al_2_O_3_ and CuO/Al_2_O_3_ nanomaterials spectra. Further investigation was carried out by XRD analysis using Kα radiation (λ = 1.5418 Å) under 40 kV voltage and an operating current of 35 mA. The results of XRD were estimated at ambient temperature using a scan rate of 0.3 s/point and 0.02° resolution. The morphological shape and size distribution were studied using scanning and transmission electron microscope.

### 2.7. Preparation of Standard NTX Solution

The standard NTX (1.0 × 10^−2^ mol L^−1^) solution was prepared by dissolving 0.341 g of pure NTX powder in 100 mL deionized water. The different working analytical solutions were prepared by serial dilution in water.

### 2.8. Sensor Construction and Membrane Composition

A conventional (NTX-TPB) coated wire sensor was constructed using a mixture of (PVC, 190 mg), electroactive material (NTX-TPB, 10 mg) and plasticizer (*o*-NPOE, 0.35 mL) in 5 mL of THF. The resulted polymeric mixture was poured into a Petri dish and kept at room temperature to evaporate slowly. The tip of the aluminium wire was polished and cleaned using deionized water, followed by acetone. The tip of the cleaned wire was immersed several times in the polymeric membrane solution (NTX-TPB) until the formation of coated membrane on its surface. To construct the modified sensor another clean Al wire was dipped three times in the polymeric solution of CuO/Al_2_O_3_ nanocomposite to produce a thin layer membrane on its surface. The sensor was allowed to dry, then dipped again in the above polymeric (NTX-TPB) solution several times until the formation of a uniform coated membrane. Both coated wire sensors were assembled as Al wire/coated membrane/test solution//Ag/AgCl reference electrode. Figure 2 demonstrated the preparation of the designed modified NTX-TPB-CuO/Al_2_O_3_ nanocomposite sensor and the potentiometric system.

### 2.9. Calibration Graphs

The linear relationship between the -logarithm NTX concentrations (mol L^−1^) and the potential difference (mV) of each conventional (NTX-TPB) and modified (NTX-TPB-CuO/Al_2_O_3_) nanocomposite was determined. The calibration graphs were plotted using 50 mL of NTX standard solutions in the concentration range 1.0 × 10^−9^–1.0 × 10^−2^ mol L^−1^ using the constructed working NTX-TPB or NTX-TPB-CuO/Al_2_O_3_ nanocomposite sensors separately in connection with a reference electrode (Ag/AgCl). The surface of the sensor should be cleaned with deionized water and dried with tissue paper prior to every measurement.

### 2.10. Optimization of Analytical Conditions

The potential response of the constructed coated wire sensors can be significantly affected by the change in the pH of the investigated solutions. The suitable pH range of conventional NTX-TPB and modified NTX-TPB-CuO/Al_2_O_3_ nanocomposite sensors was determined using 1.0 × 10^−4^ mol L^−1^ NTX solution. The pH of the tested sample was acidified using 0.1 mol L^−1^ of hydrochloric acid. The potential response of the working sensor was measured after increasing the pH value using 0.1 mol L^−1^ sodium hydroxide solution. The pH graphs were constructed using the change in potential against the pH.

A separate solution method [50] was used to monitor the selectivity of the investigated NTX sensors. Briefly, the selectivity coefficient of each sensor towards NTX and some foreign substances and additives such as (Povidone, hydroxypropyl methylcellulose, lactose monohydrate, magnesium stearate, microcrystalline cellulose, polyethylene glycol, polysorbate 80, tryptophan, lysine, and glycine) was determined using 1.0 × 10^−3^ mol L^−1^ of NTX solution. The selectivity coefficient of conventional NTX-TPB and modified NTX-TPB-CuO/Al_2_O_3_ nanocomposite was calculated using the following equation:Log K^pot^ = (E_2_ − E_1_)/S + log [Drug] − Log [B^z+^]^1/z^(1)
where K^pot^ (selectivity coefficient), E_1_ (electrode potential) of 1.0 × 10^−3^ mol L^−1^ NTX, E_2_ (electrode potential) of 1.0 × 10^−3^ mol L^−1^ of interfering species, B^z+^ (interfering species), and S (slope) of the calibration curve, respectively. The response time was tested by recording the dynamic potential response of the investigated drug, using NTX concentration range of 1.0 × 10^−9^–1.0 × 10^−2^ mol L^−1^.

### 2.11. Estimation of Naltrexone Hydrochloride^®^ Tablets

The content of ten naltrexone hydrochloride^®^ tablets (50 mg/tablet) was pulverized and mixed well. An accurate quantity of 0.341 g was dissolved in 50 mL deionized water, then centrifuged at 1500 rpm for 5 min and filtered to remove the co-additive materials. The clear solution was completed with deionized water to be 100 mL, the resulting NTX solution (1.0 × 10^−2^ mol L^−1^) was diluted with the same solvent to prepare the working analytical samples in the range of 1.0 × 10^−5^–1.0 × 10^−2^ and 1.0 × 10^−9^–1.0 × 10^−2^ mol L^−1^. The designed NTX-TPB and modified NTX-TPB-CuO/Al_2_O_3_ nanocomposite sensors were separately utilized to quantify the investigated drug in its commercial tablets.

## 3. Results and Discussion

### 3.1. Characterization of CuO/Al_2_O_3_ Nanocomposite

Various spectroscopic techniques such as UV-Vis, FT-IR, XRD, and EDS were exploited to characterize the synthesized CuONPs/Al_2_O_3_NPs nanocomposite. One of the most suitable and useful methods for primary confirmation of shape, size and stability of the engineered nanoparticles in their aqueous suspensions is the UV-Vis method. The recorded spectrum of CuONPs/Al_2_O_3_NPs displayed two significant broad absorption peaks at 280 and 400 nm for CuO and Al_2_O_3_ nanoparticles, respectively (Figure 3). The band gaps of the as-synthesized metal oxide nanoparticles were calculated from the formula:Eg = hυ = hc/λ(2)
where “h is Planck’s constant, c is the velocity of light, and λ is the wavelength.” The optical band gap energy of CuONPs and Al_2_O_3_NPs were estimated to be 2.85 eV and 3.54 eV, respectively [51,52].

The FT-IR spectra of the pre-synthesized CuONPs, Al_2_O_3_NPs, and CuO/Al_2_O_3_ nanocomposite were performed in the range in range of 400–4000 cm^−1^. The FT-IR spectrum of CuONPs (Figure 4a) showed different absorption bands at 3424.25 cm^−1^ (O-H stretching vibration), 1632.22 cm^−1^ (O-H bending vibration of absorbed water), 1466.58 cm^−1^ and 1112.15 cm^−1^ (CO_2_ of the surrounding atmosphere), and 590 cm^−1^ and 511 cm^−1^ (Cu-O stretching bond formation). The obtained results are in agreement with those previously reported [53]. Figure 4b, described the FT-IR spectrum of Al_2_O_3_NPs. Broad and weak absorption bands appeared at 3499.11 cm^−1^ and 1644.25 cm^−1^ were due to stretching and bending O-H vibration of absorbed water, respectively; the band observed at 1369.12 cm^−1^ was corresponding to stretching vibration of Al-OH bond, 1087.00 cm^−1^ assigned to be corresponding to strong C-O stretching vibration. The peaks at 846.14 cm^−1^ and 607.15 cm^−1^ correspond to the Al-O bond [52]. In the CuO/Al_2_O_3_ nanocomposite FT-IR spectrum, different absorption vibration peaks were recorded at 3432.42 cm^−1^ (O-H stretching vibration), 2353.69 cm^−1^ (O=C=O of the surrounding atmosphere and 1633.56 cm^−1^ (O-H vibration mode of water). The appearance of two stretching vibration peaks at 520.15 and 608.14 cm^−1^ indicated the formation of CuO/Al_2_O_3_ nanocomposite (Figure 4c).

The XRD analysis were performed to study the crystalline structure of the synthesized CuONPs, Al_2_O_3_NPs and CuO/Al_2_O_3_ nanocomposite. The XRD pattern of CuONPs showed various characteristic peaks at 2Ɵ = 32.5° (1 1 0), 35.7° (0 0 2), 38.9° (1 1 1), 47.8° (1 1 2), 52.4° (2 0 2), 63.8° (1 1 3), 86.4° (2 2 0), 76.5° (3 1 1) plane orientation of CuO which was verified from the Standard Joint Committee on Powder Diffraction Standards (JCPDS 80-1268). The recorded results from peak positions confirmed the monoclinic structure of the CuONPs and no other phases were noticed, revealing high purity of the formed CuONPs (Figure 5a). The XRD pattern of Al_2_O_3_ nanoparticles showed that the sample was in nanorods crystalline shape and exhibited different peaks at 32.65° (2 2 0), 35.14° (3 1 1), 37.26° (2 2 2), 43.58° (4 0 0), 54.22° (4 2 2), 59.84° (4 4 0), 74.89° (6 2 0). All observed diffraction peaks can be matched to those of bulk α-Al_2_O_3_, with cell constant (α = 8.395 Å), which is in acceptance with the value of (JCPDS file No.10-173). No other impurity phases can be detected, indicating high purity of the synthesized Al_2_O_3_ nanoparticles (Figure 5b). Very similar diffraction peaks were observed in the XRD pattern of CuO/Al_2_O_3_ nanocomposite. Therefore, the above diffraction peaks can be used to verify the formation of CuO/Al_2_O_3_ nanocomposite (Figure 5c). The prominent peak in the XRD pattern was used to calculate the average crystallite size using a Scherrer formula:D = 0.9λ/ β cos θ(3)
where λ, θ and β are the X-ray wavelength, Bragg diffraction angle, and the Full width at half maximum of the XRD peak appearing at a diffraction angle θ, respectively. The average crystallite size of the synthesized CuONPs and Al_2_O_3_NPs were evaluated to be 17.09 ± 1.1 and 27.61 ± 3.1 nm, respectively.

The dislocation density (δ), which describes the quantity of defects in the sample is known as the length of dislocation lines per unit volume of the crystal and is calculated using the following equation [54].
(4)$$\delta \;=\;1/D^{2}$$
where D is the crystallite size. The dislocation density of CuONPs and Al_2_O_3_NPs at room temperature was found to be 1.42 × 10^−3^ and 1.31 × 10^−3^ (nm)^−2^, respectively.

The bond length of Cu-O and Al-O was calculated from the equation [55].
(5)$$L=\sqrt{(\frac{a^{2}}{3}+\left(\frac{1}{2}-u{)}^{2}c^{2}\right)}$$
where *u* is the positional parameter in the wurtzite structure and is a measure of the amount by which each atom is displaced according to the next along the ‘*c*’ axis. ‘*u*’ is given by the equation:(6)$$u=\frac{a^{2}}{3c^{2}}+0.25\;$$

The *u* value increases with the decreases in *c*/*a* ratio this for tetrahedral distances Remaining constant via a distortion of tetrahedral angels. The calculated bond length of Cu-O and Al-O was found to be 1.928 and 1.869 Å, respectively. The calculated bond length is an agreement with the bond length of Cu-O and Al-O in the unit cell [56,57].

Further microscopic studies were performed using TEM and SEM to confirm the size, shape and surface morphology of CuO/Al_2_O_3_ nanocomposite according to the as-prepared CuONPs and Al_2_O_3_NPs. The obtained images of TEM (Figure 6a–c) and SEM (Figure 6d–f) confirmed the uniform distribution of the formed spherical and rode shapes CuONPs and Al_2_O_3_NPs, respectively. However, the surface morphology of CuO/Al_2_O_3_ nanocomposite showed highly aggregated crystals with particle sizes ranging from 80–100 nm.

SEM equipped with an EDX spectroscopy was used to evaluate the elemental composition of Cu, Al in the prepared CuO/Al_2_O_3_ nanocomposite with respect to the EDX profiles of CuONPs and Al_2_O_3_NPs. The obtained EDX profiles of CuONPs and Al_2_O_3_NPs showed that the percentage elemental content of Cu, Al, and O was 75.56% (Cu) and 24.44% (O) for CuONPs, 69.88% (Al), and 30.12% (O) for Al_2_O_3_NPs, with a maximum peak intensity 1.5 and 1.2 keV for Cu and Al, respectively (Figure 7a,b). Whereas the recorded EDX profile of CuO/Al_2_O_3_ nanocomposite exerts 57.46% Cu, 31.97% Al, and 10.57% O, which suggested the complete reduction of Cu an Al and the high purity of the synthesized CuO/Al_2_O_3_ nanocomposite (Figure 7c).

### 3.2. Characteristics of the Constructed Sensors

NTX reacts with TPB to produce a very stable complex NTX-TPB, which is soluble in THF. The conventional NTX-TPB and modified coated wire NTX-TPB-CuO/Al_2_O_3_NPs nanocomposite sensors were designed by mixing the electroactive materials (ion pairs) with PVC and a solvent mediator (*o*-NPOE) using THF. The current study suggested the use of a high dielectric constant (*o*-NPOE, ε=24) serves as a fluidizer allowing the homogenous dissolution of the electroactive materials and facilitating its mobility and diffusion through the polymeric matrix of the membrane. Also, it enhances the selectivity coefficient of the sensor by offering a suitable mechanical property for the coated membrane [53]. Table 1 reported the effective potential response and the potentiometric behavior of the constructed NTX-TPB and NTX-TPB-CuO/Al_2_O_3_NPs nanocomposite sensors. The obtained data showed that the designed sensors exhibited Nernstian behavior with slopes of E_mV_ = (52.1 ± 0.2) log [NTX] + 406.6 and E_mV_ = (58.25 ± 0.3) log [NTX] + 754.25 with linearity ranges 1.0 × 10^−5^–1.0 × 10^−2^ mol L^−1^ (r^2^ = 0.9995) and 1.0 × 10^−9^–1.0 × 10^−2^ mol L^−1^ (r^2^ = 0.9999) for the above-mentioned sensors, respectively (Figure 8a,b). The addition of CuO/Al_2_O_3_ nanocomposite to modify the conventional NTX-TPB sensor enhanced the potential response of the designed sensor (NTX-TPB- CuO/Al_2_O_3_ nanocomposite) to a wider linear detection range with high sensitivity towards the detection of NTX solution. These results can be attributed to the large surface area of the added nanoparticles which increased the surface conductivity of the designed modified sensor. Additionally, the higher detection results observed by using the modified sensor could be due to the high dielectric permittivity value of CuONPs (≈10^4^) and Al_2_O_3_NPs (≈7.8–11.1) at ambient temperature [58,59]. The dynamic response of the designed NTX-TPB conventional and modified NTX-TPB-CuO/Al_2_O_3_NPs nanocomposite sensors were investigated under optimized experimental conditions to detect the difference between the instant potential time and its steady-state value (1 mV) [60]. The recorded dynamic responses were found to be 70 s and 50 s for the above-mentioned conventional and modified sensors, respectively. It was observed that the modified sensor with metal oxide nanocomposite exhibited a fastness response with high mechanical stability more than the conventional type. The modification of the membrane with high surface area to volume ratio metal oxides nanocomposite and new unique physicochemical properties enhances the electrical conductivity of the modified sensor towards the detection of the tested analyte in the sample. Moreover, the remarkable electrical and exceptional capacity characteristics of metal oxides nanocomposite such as the high charge transfer resulting at the nanomaterial interfaces are of paramount importance when the nanomaterials are employed as transducing materials in sensing applications [61].

The hydrogen ion concentration can greatly affect the potential response of the membrane sensor. Thus, it is very necessary to determine the suitable pH range where the potential response of the coated membrane sensor is not affected by hydrogen ions. The obtained results showed that the above-mentioned NTX-TPB and NTX-TPB-CuO/Al_2_O_3_NPs nanocomposite sensors are independent in the pH range 2–5 and NTX can be simply estimated using the designed sensors within this pH range (Figure 9). It was observed that at high [H^+^] in acidic medium (pH < 2), the protonated ion-pair complex was formed, and the potential readings of the sensors were slightly increased due to the poor responsiveness to NTX ions, whereas at high [OH^-^] in alkaline medium (pH > 5) the potential readings were gradually decreased as a result of a competition between NTX ions and OH- ions and hence reduces the interaction between the investigating drug ions and the sites of ion-pair on the sensor membrane [62].

A separate solution method [50] was applied to evaluate the interference effect of some foreign substances on the selectivity coefficient of the constructed NTX sensors using 1.0 × 10^−3^ mol L^−1^ solution. The modified NTX-TPB-CuO/Al_2_O_3_ nanocomposite sensors exhibited excellent selectivity. The unique physicochemical features of the synthesized CuO/Al_2_O_3_NPs and their large surface area increase the conductivity of the designed modified sensor and hence elevate its selectivity towards the detected NTX ions. Moreover, the NTX coated membrane selectivity is refer to the free energy transfer of ions (NTX^+^) generated between the active sites in the membrane and the working solution. No interferences were noticed by the tested co-additives and amino acids. Thus, excellent selectivity and good tolerance were accomplished by using the modified NTX sensor for the determination of NTX (Table 2).

### 3.3. Estimation of NTX in Bulk Form

The suggested conventional NTX-TPB and NTX-TPB-CuO/Al_2_O_3_ sensors were used to estimate NTX in its bulk form and the obtained results were calculated as percentage recoveries of 98.33 ± 0.9% and 99.81 ± 0.2% for the above-mentioned sensors, respectively (Table 3). The high dynamic response of the modified sensor can be due to the advanced conductivity and dielectric permittivity properties of the used CuONPs (≈10^4^) and Al_2_O_3_NPs (≈7.8–11.1), which provide excellent sensitivity and selectivity towards NTX solution.

### 3.4. Validation Study

The International Council for Harmonization of Technical Requirements for Pharmaceuticals (ICH) guidelines [49] were employed to ensure the validity and suitability of the suggested potentiometric method. Linear relationships in the concentration ranges 1.0 × 10^−5^–1.0 × 10^−2^ and 1.0 × 10^−9^–1.0 × 10^−2^ mol L^−1^ with least square regression equations E_mV_ = (52.1 ± 0.2) log [NTX] + 406.6 (r^2^= 0.9995) and E_mV_ = (58.25 ± 0.3) log [NTX] + 754.25 (r^2^ = 0.9999) and low detection limits of 5.0 × 10^−6^ and 5.0 × 10^−10^ mol L^−1^ for the conventional and modified NTX sensors, respectively.

The accuracy of the suggested potentiometric method was studied using nine authentic NTX concentrations and the results were expressed as mean percentage recoveries as 98.73 ± 1.09% and 99.72 ± 0.4% for NTX-TPB and NTX-TPB-CuO/Al_2_O_3_ nanocomposite, respectively (Table 4). The precision was also studied using intra-day and inter-day assays. The recorded results were expressed as a percentage relative standard deviation (%RSD) as 0.3% and 0.2% for the two constructed NTX sensors, respectively (Table 5). The system robustness was evaluated by performing a slight variation in method parameter by changing the pH of working solutions to 5 ± 0.5. The calculated percentage recoveries were 98.45 ± 0.7% and 99.63 ± 0.3% for the conventional and modified NTX coated wire sensors (Table 1). Another experiment was carried out to ensure the ruggedness of the suggested potentiometric method by performing the analysis using another pH-meter (Metrohm model-744) in the different laboratories and different operators. The mean percentage recoveries were found to be 98.59 ± 0.6% and 99.68 ± 0.4% for the above NTX sensors, respectively (Table 1). The outcomes of method validation revealed a good agreement with those resulting from the suggested method and no remarkable differences was noticed.

### 3.5. Analysis of NTX in Naltrexone Hydrochloride^®^ Tablets

The investigated NTX was determined in its commercial tablets Naltrexone hydrochloride^®^ (50 mg/tablet) using the constructed NTX-TPB and NTX-TPB-CuO/Al_2_O_3_ nanocomposite. The potential readings of the working solutions 1.0 × 10^−5^–1.0 × 10^−2^ and 1.0 × 10^−9^–1.0 × 10^−2^ mol L^−1^ were measured and the percentage recoveries of NTX were estimated from the regression equations. The recorded results were calculated as 99.05 ± 0.5% and 99.70 ± 0.2% for the above-designed sensors. The obtained results were compared with the reported method by Ganjali et al. [63] using the t-student’s test and F-test [64] and showed excellent sensitivity and selectivity of the designed sensor for the determination of NTX (Table 6). The dielectric constant is a critical factor that evaluate the capability of the materials to store charges [65]. Metal oxides with high dielectric constant are commonly used in electronics and sensors. As they do not permit the flow of charges through them, they allow for exerting electrostatic fields and storing charges [66]. The combination of metal oxides nanoparticles with polymeric matrix in nanocomposites could effectively improve the electrical, optical and conductive properties of the modified sensor. These properties are much sensitive to changes in the particles shape and size. As previously reported the nanoparticles themselves could serve as conductive junctions between the polymeric chains that resulted in as increase of electrical conductance of the composites [67]. Additionally, the modification of the sensor with nanocomposite containing metal oxides with high surface area to volume ratios and possess new physicochemical properties enhanced the charge transfer and the electrical conductivity of the sensor towards the interaction with the target analyte in the test solution and resultingly improve sensitivity of the sensor detection [68].

A comparative study was carried out to compare the efficiency of the designed modified NTX-TPB-CuO/Al_2_O_3_ nanocomposite sensor with the previously constructed sensors [63,69,70]. The comparative results revealed high sensitivity of the modified sensor towards the determination of NTX with a wide detection range of 1.0 × 10^−9^–1.0 × 10^−2^ and LOD 5.0 × 10^−6^ mol L^−1^ than the reported (Table 7). The selection of nanostructured materials and sensor design method is the most important factor for achieving ultrasensitive sensor with desired characteristics. The shape and size of the nanoparticles used governs the surface to volume ratio, which is a crucial factor to enhance the interface reactions on the overall nanomaterial’s electrical conductivity. The nanoscale morphology will not only influence the sensitivity of the sensor but also affect the dynamic response of the sensor and the long-term stability of the sensor due to the high chemical stability of these nanomaterials. The electrical conductivity of the fabricated sensors using metal oxide nanocomposite might also based on the molecular structure and the polymeric medium such as the crystallinity and long chain polymer [71].

## 4. Conclusions

The present study has described a successfully designed simple and ultrasensitive modified NTX-TPB-CuO/Al_2_O_3_ nanocomposite potentiometric sensor for the determination of NTX in authentic powder and commercial formulations. The designed modified sensor exhibited a large surface area to volume ratio which granted excellent sensitivity for the detection of NTX with linear relationships in the concentration ranges 1.0 × 10^−5^–1.0 × 10^−2^ and 1.0 × 10^−9^–1.0 × 10^−2^ mol L^−1^ with least square regression equations E_mV_ = (52.1 ± 0.2) log [NTX] + 406.6 (r^2^ = 0.9995) and E_mV_ = (58.25 ± 0.3) log [NTX] + 754.25 (r^2^ = 0.9999) and low detection limits of 5.0 × 10^−6^ and 5.0 × 10^−10^ mol L^−1^ for the conventional and modified NTX sensors, respectively. Outcomes of the suggested method were evaluated statistically and matched with those of previously addressed sensors. It was noticed that the designed modified NTX-TPB-CuO/Al_2_O_3_ nanocomposite showed a more extraordinary potential response than the conventional type. Furthermore, covering the surface of the sensor by a modified layer of metal oxide nanocomposite polymeric membrane increased the electroconductivity of this sensor and improved the quantification of the tested drug in its tablets with mean percentage recovery 99.70 ± 0.2% for the above-modified sensor, revealing high sensitivity and selectivity. Therefore, the use of metal oxide nanocomposite in the construction of polymeric sensors opens up a promising area in developing novel modified potentiometric sensors.

## Figures and Tables

**Figure 1 polymers-13-04459-f001:**
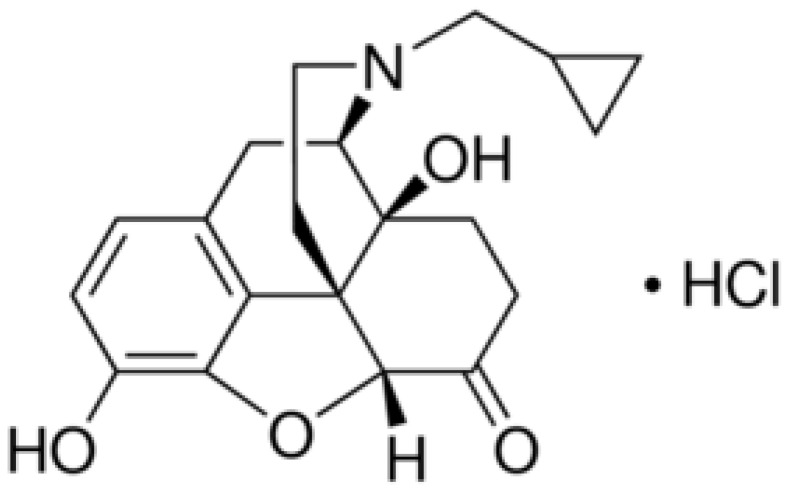
Structural formula of Naltrexone hydrochloride.

**Figure 2 polymers-13-04459-f002:**
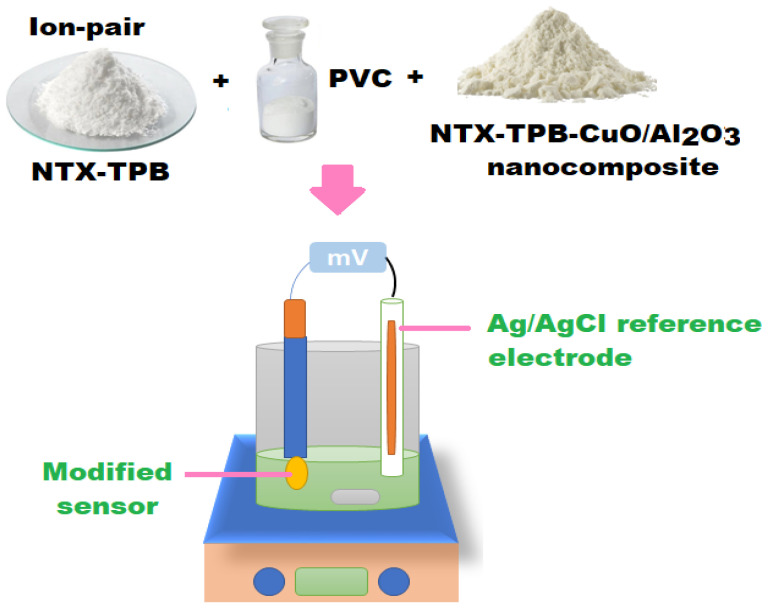
Schematic illustration for the preparation of the designed modified NTX-TPB-CuO/Al_2_O_3_ nanocomposite sensor and the potentiometric system.

**Figure 3 polymers-13-04459-f003:**
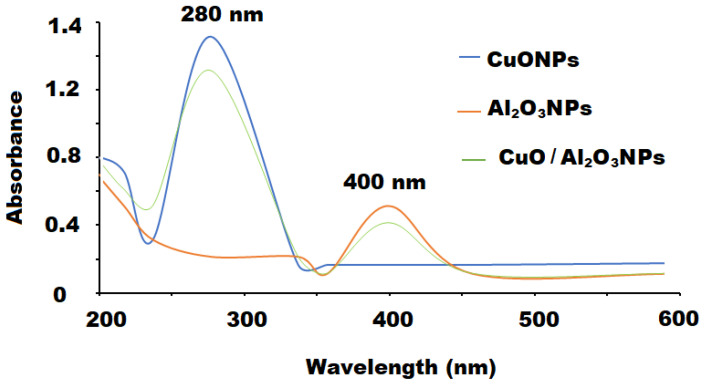
UV-vis spectra of CuONPs (λ_max_ = 280 nm), Al_2_O_3_NPs (λ_max_ = 400 nm) and CuO/Al_2_O_3_ nanocomposite measured at absorption wavelenth ranged between 200–500 nm.

**Figure 4 polymers-13-04459-f004:**
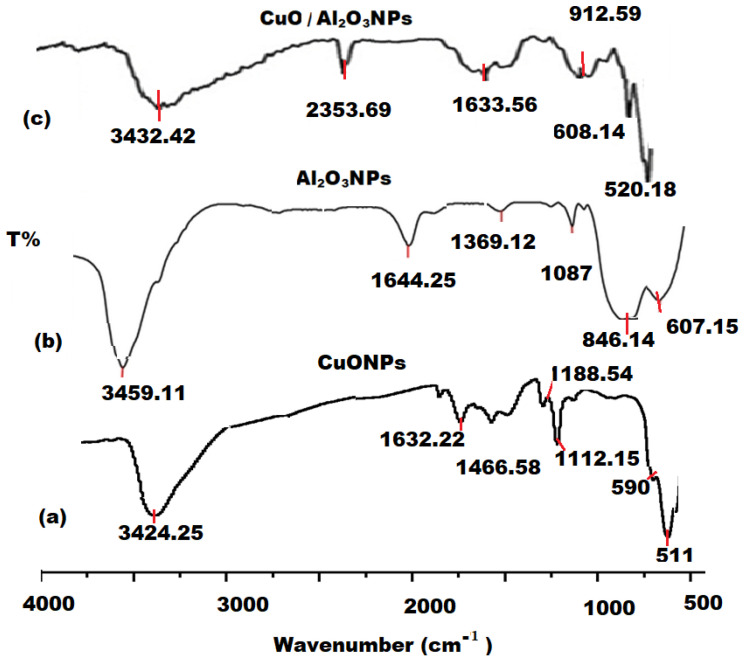
Fourier-Transform Infrared (FT-IR) spectra of the as-synthesized (**a**) CuONPs, (**b**) Al_2_O_3_NPs and (**c**) CuO/Al_2_O_3_ nanocomposite measured at wavenumber range 4000–500 cm^−1^.

**Figure 5 polymers-13-04459-f005:**
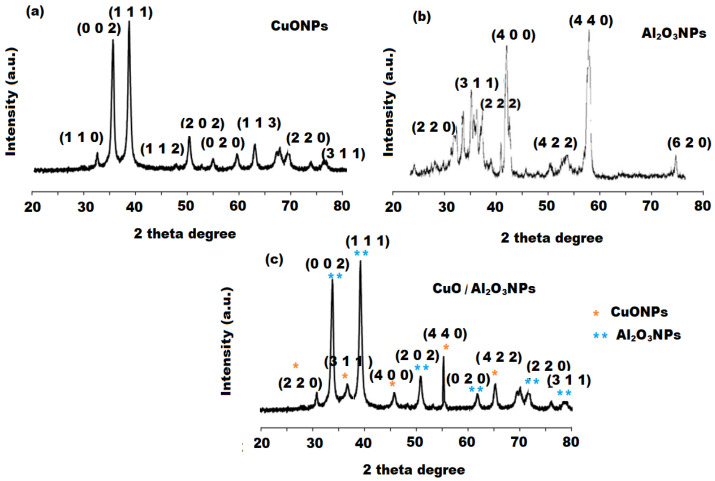
XRD spectra of (**a**) CuONPs, (**b**) Al_2_O_3_NPs and (**c**) CuO/Al_2_O_3_ nanocomposite.

**Figure 6 polymers-13-04459-f006:**
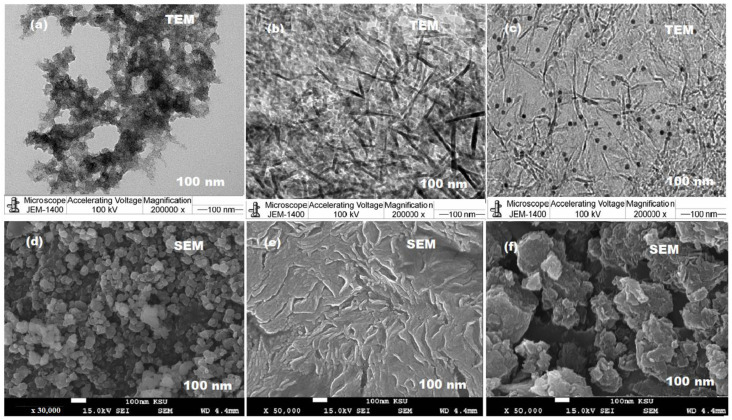
(**a**–**c**) TEM and (**d**–**f**) SEM images of CuONPs, Al_2_O_3_NPs and CuO/Al_2_O_3_ nanocomposite, respectively.

**Figure 7 polymers-13-04459-f007:**
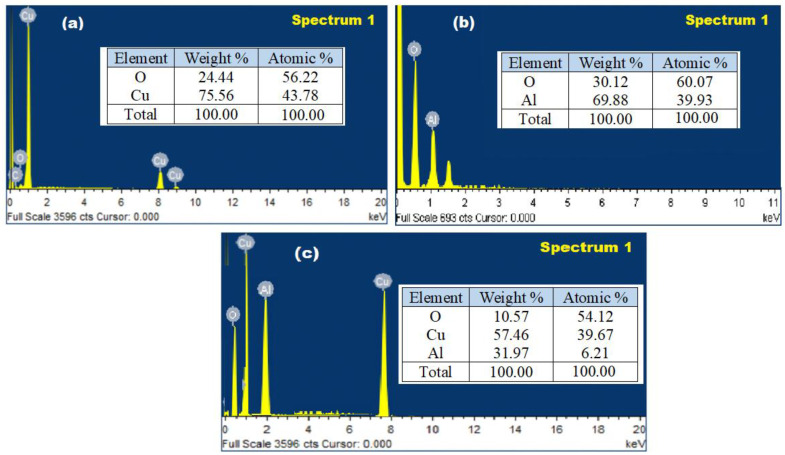
Energy Dispersive X-Ray (EDX) analysis of (**a**) CuONPs, (**b**) Al_2_O_3_NPs and (**c**) CuO/Al_2_O_3_ nanocomposite.

**Figure 8 polymers-13-04459-f008:**
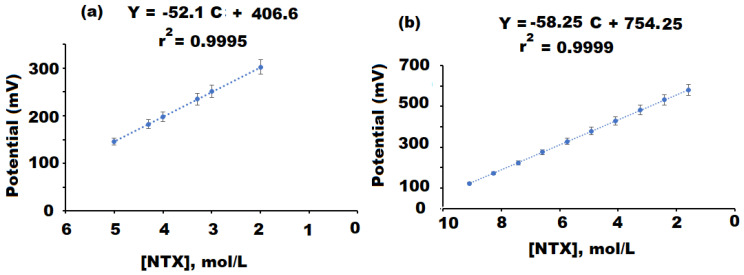
Linear relationships of (**a**) Conventional NTX-TPB and (**b**) NTX-TPB-CuO/Al_2_O_3_ nanocomposite coated wire sensors.

**Figure 9 polymers-13-04459-f009:**
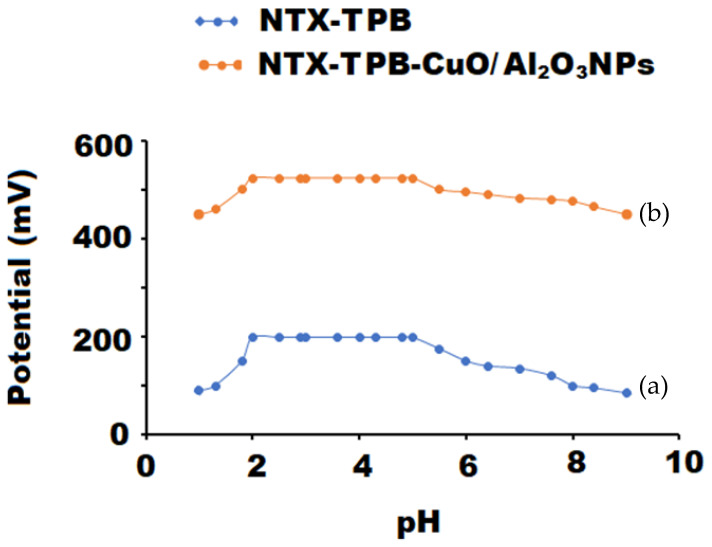
pH range of each constructed Conventional (**a**) NTX-TPB and (**b**) NTX-TPB-CuO/Al_2_O_3_ nanocomposite coated wire sensors using 1.0 × 10^−4^ mol L^−1^ of NTX solution.

**Table 1 polymers-13-04459-t001:** Critical response features of designed conventional coated wire naltrexone -tetraphenyl borate (NTX-TPB) and modified NTX-TPB-CuO/Al_2_O_3_ nanocomposite sensors.

Parameter	Conventional Coated Wire NTX-TPB Sensor	Modified NTX-TPB-CuO/Al_2_O_3_ Nanocomposite Sensor
Linear conc. (mol L^−1^)	1.0 × 10^−6^–1.0 × 10^−2^	1.0 × 10^−11^–1.0 × 10^−2^
Regression equation	E_mV_ = (52.1 ± 0.2) log [NTX] + 406.6	E_mV_ = (58.25 ± 0.3) log [NTX] + 754.25
Correlation Coefficient (r)	0.9995	0.9999
Slope (mV. Decade^−1^)	52.1 ± 0.2	58.25 ± 0.3
Intercept	406.6	754.25
LOD	5.0 × 10^−6^	5.0 × 10^−10^
Suitable pH range	2–5	2–5
Temperature (°C)	25	25
Lifetime (day)	30	60
Accuracy (%)	98.73 ± 1.09	99.72 ± 0.4
Robustness	98.45 ± 0.7	99.63 ± 0.3
Ruggedness	98.59 ± 0.4	99.68 ± 0.4

**Table 2 polymers-13-04459-t002:** The tolerable values of some foreign substances and co-formulated materials measured by the designed conventional coated wire NTX-TPB and modified NTX-TPB-CuO/Al_2_O_3_ nanocomposite sensors using separate solution method.

Interferences	Conventional Coated Wire NTX-TPB Sensor (K^Pot^_NTX_^+^)	Modified NTX-TPB-CuO/Al_2_O_3_ Nanocomposite Coated Wire Sensor (K^Pot^_NTX_^+^)
Povidone	5.6 × 10^−3^	1.5 × 10^−5^
Hydroxypropyl methylcellulose	1.9 × 10^−3^	5.6 × 10^−4^
Lactose monohydrate	4.2 × 10^−3^	2.8 × 10^−5^
Magnesium stearate	9.4 × 10^−3^	3.9 × 10^−5^
Microcrystalline cellulose	4.9 × 10^−3^	1.7 × 10^−4^
Polyethylene glycol	6.3 × 10^−3^	3.3 × 10^−5^
Polysorbate 80	1.2 × 10^−3^	6.9 × 10^−5^
Tryptophan	5.6 × 10^−3^	2.4 × 10^−4^
Lysine	5.3 × 10^−3^	5.0 × 10^−5^
Glycine	3.3 × 10^−3^	3.6 × 10^−5^

**Table 3 polymers-13-04459-t003:** The calculated resulted of pure NTX estimation using the designed conventional coated wire NTX-TPB and modified NTX-TPB-CuO/Al_2_O_3_ nanocomposite sensors.

	Conventional Coated Wire NTX-TPB Sensor		Modified NTX-TPB-CuO/Al_2_O_3_ Nanocomposite Coated Wire Sensor	
	Test Sample −log [NTX] mol L^−1^	% Recovery	Test Sample −log [NTX] mol L^−1^	% Recovery
Statistical analysis	6.0	99.5	9.0	99.9
	5.3	98.9	8.0	99.6
	5.0	99.0	7.0	100.0
	4.0	97.3	6.0	99.7
	3.0	97.3	5.0	99.6
	2.0	98.0	4.0	100.0
			3.0	99.7
			2.0	100.0
Mean ± SD	98.33 ± 0.9		99.81 ± 0.2	
n	6		8	
Variance	0.81		0.04	
%SE *	0.37		0.07	
%RSD	0.92		0.20	

* %SE (%Error) = %RSD/$\sqrt{n}$.

**Table 4 polymers-13-04459-t004:** The accuracy study for NTX estimation using the designed conventional coated wire NTX-TPB and modified NTX-TPB-CuO/Al_2_O_3_ nanocomposite sensors.

	Conventional Coated Wire NTX-TPB Sensor		Modified NTX-TPB-CuO/Al_2_O_3_ Nanocomposite Coated Wire Sensor	
	Test Sample −log [NTX] mol L^−1^	% Recovery	Test Sample −log [NTX] mol L^−1^	% Recovery
Statistical Analysis	6.0	99.9	9.0	99.9
	5.3	97.9	8.3	99.7
	5.0	99.5	8.0	99.9
	4.3	98.3	7.0	99.6
	4.0	97.3	6.0	100.0
	3.3	98.9	5.0	99.7
	3.0	99.7	4.0	98.8
	2.3	97.2	3.0	100.0
	2.0	99.9	2.0	99.9
Mean ± SD	98.73 ± 1.09		99.72 ± 0.4	
n	9		9	
Variance	1.18		0.16	
%SE *	0.36		0.13	
%RSD	1.10		0.40	

* %SE (%Error) = %RSD/$\sqrt{n}$.

**Table 5 polymers-13-04459-t005:** Intermediate precision assay of the designed modified NTX-TPB-CuO/Al_2_O_3_ nanocomposite coated wire sensors.

	Modified NTX-TPB-CuO/Al_2_O_3_ Nanocomposite Coated Wire Sensor					
	Intra-Day Assay			Inter-Day Assay		
	Sample −log [NTX] mol L^−1^	Found −log [NTX] mol L^−1^	% Recovery	Sample −log [NTX] mol L^−1^	Found −log [NTX] mol L^−1^	% Recovery
Statistical Analysis	11.0	10.99	99.9	11.0	11.0	100.0
	7.0	7.0	100.0	7.0	6.98	99.7
	2.0	1.99	99.5	2.0	2.01	100.2
Mean ± SD	99.8 ± 0.3			99.9 ± 0.2		
n	3			3		
Variance	0.09			0.04		
%SE *	0.17			0.11		
%RSD	0.30			0.20		

* %SE (%Error) = %RSD/$\sqrt{n}$.

**Table 6 polymers-13-04459-t006:** The calculated resulted from the quantification of NTX in Naltrexone hydrochloride^®^ 50 mg/tablet using the designed conventional coated wire NTX-TPB and modified NTX-TPB-CuO/Al_2_O_3_ nanocomposite sensors.

	Conventional Coated Wire NTX-TPB Sensor		Modified NTX-TPB-CuO/Al_2_O_3_ Nanocomposite Coated Wire Sensor		Reported Method [63]
	Test Sample −log [NTX] mol L^−1^	% Recovery	Test Sample −log [NTX] mol L^−1^	% Recovery	
Statistical analysis	6.0	99.8	11.0	100.00	
	5.3	99.6	9.0	99.9	
	5.0	98.8	7.0	99.8	99.52 ± 0.3
	4.0	98.5	5.0	99.6	6
	3.0	98.7	3.0	99.4	0.09
	2.0	98.9	2.0	99.5	0.12
					0.30
Mean ± SD	99.05 ± 0.5		99.70 ± 0.2		
n	6		6		
Variance	0.25		0.04		
%SE *	0.20		0.08		
%RSD	0.50		0.20		
*t*-student test	2.015 (2.228) *		1.248(2.228) *		
F-test	2.77 (5.05) *		2.25(5.05) *		

* Tabulated values of t-student test and F-test at *p* < 0.05 [64].

**Table 7 polymers-13-04459-t007:** Comparative study between the suggested modified NTX-TPB-CuO/Al_2_O_3_ nanocomposite sensor and the previously potentiometric reported method.

No.	Ion-Pair Complex	Concentration Range (mol L^−1^)	LOD (mol L^−1^)	Reference
1.	NTX-tetraphenylborate	1.0 × 10^−5^–1.0 × 10^−2^	8.0 × 10^−6^	[63]
2.	NTX-tetraphenylborate	1.0 × 10^−5^–1.0 × 10^−3^	5.0 × 10^−6^	[69]
3.	NTX-tetrakis-4 chlorophenyl borate	5.8 × 10^−6^–1.0 × 10^−2^	5.0 × 10^−6^	[70]
4.	NTX-tetraphenylbotate-CuO/Al_2_O_3_ nanocomposite	1.0 × 10^−9^–1.0 × 10^−2^	5.0 × 10^−10^	Current study

## Data Availability

All data resulted from this study was presented in the text.

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
