# Peer review of "New Construction of Functionalized CuO/Al_2_O_3_ Nanocomposite-Based Polymeric Sensor for Potentiometric Estimation of Naltrexone Hydrochloride in Commercial Formulations"

_polymers, 2021, doi:10.3390/polym13244459_

Round 1
Reviewer 1 Report
1-Re arrange the paragraphs and remove the unreasonable distances ( lines: 162, 221, 233, 266, 268, 298, 343, 354, ......etc)
2- Numbering the used equations in the manuscript
3-The text is not sufficiently clear and the use of English should be improved
4- Please cite the previously reported CuO/Al2O3 NPs for sensing applications
5- "D" in Scherrer formula represent the crysatallite size, the particle size or diameter can be estimated from SEM and TEM and comapered with "D" if D= particle size then each particle contains only one crytsallite.
6- You can calculate other structural parameters such as dislocation density, texture coefficient, bond length, ......etc
7- L393-395: "The high sensitivity of the modified sensor was due to the dielectric constant of CuO and Al2O3 nanoparticles which improve the conductivity of the
sensor and increase the response to the measured drug." Please provide more information to clarify the relation between the dielectric constant and sensor sensitivity
8- It is important to correlate the structural and morphological properties with the sensor performance
9- Highlight the novelty of this work at the end of the introduction or at the end of the results and discussion
10- Provide the main key finds in the conclusion
Author Response
Reviewer 1:
Thank you very much for your valuable comments. The manuscript has been carefully revised and all comments have been corrected point by point and the correction were highlighted using red color.
1-Re arrange the paragraphs and remove the unreasonable distances (lines: 162, 221, 233, 266, 268, 298, 343, 354, ......etc)
Response: All unreasonable distances have been removed from the suggested lines.
2- Numbering the used equations in the manuscript
Response: The used equations in the manuscript has been numbered
3-The text is not sufficiently clear and the use of English should be improved
Response: English improvement has been performed throughout the whole text and the corrections highlighted with red colour.
4- Please cite the previously reported CuO/Al2O3NPs for sensing applications
Response: The previously reported CuO/Al2O3NPs for sensing applications have been cited in the introduction section (Ref 20-22) and all reference numbers have been corrected.
5- "D" in Scherrer formula represent the crystallite size, the particle size or diameter can be estimated from SEM and TEM and compered with "D" if D= particle size then each particle contains only one crystallite.
Response: The crystallite size was calculated again and the corrected values have been inserted in the text (34.87±1.1 and 36.13±3.1 nm) for CuO and Al2O3, respectively.
6- You can calculate other structural parameters such as dislocation density, texture coefficient, bond length, …. etc
Response: Other parameters have been calculated such as delocalization densitybond length and added in the text
7- L393-395: "The high sensitivity of the modified sensor was due to the dielectric constant of CuO and Al2O3 nanoparticles which improve the conductivity of the sensor and increase the response to the measured drug. "Please provide more information to clarify the relation between the dielectric constant and sensor sensitivity.
Response: More information to clarify the between the dielectric constant and sensor sensitivity was added in the text and four new references were incorporated in this paragraph.
The dielectric constant is a critical factor that evaluate the capability of the materials to store charges [61]. Metal oxides with high dielectric constant are commonly used in electronics and sensors. As they do not permit the flow of charges through them, they allow exerting electrostatic field and hence storing charges [62]. The combination of metal oxides nanoparticles with polymeric matrix in nanocomposites could effectively improve the electrical, optical and conductive properties of the modified sensor. These properties are much sensitive to changes in the particles shape and size. As previously reported [63] the nanoparticles themselves could serve as conductive junctions between the polymeric chains that resulted in as increase of electrical conductance of the composites. Additionally, the modification of the sensor with nanocomposite containing metal oxides with high surface area to volume ratios and possess new physicochemical properties enhanced the charge transfer and the electrical conductivity of the sensor towards the interaction with the target analyte in the test solution and hence improve the sensitivity of the sensor detection [64].
8- It is important to correlate the structural and morphological properties with the sensor performance the structural and morphological properties with the sensor performance
Response: The correlation between the structural and morphological properties with the sensor performance has been included in the text.
The selection of metal oxide nanostructured materials and sensor design method is the most important factor for achieving ultrasensitive sensor with desired characteristics. The shape and size of the nanoparticles used governs the surface to volume ratio, that is a crucial factor to enhance the interface reactions on the overall nanomaterial’s electrical conductivity. The nanoscale morphology will not only influence the sensitivity of the sensor but also affect the dynamic response of the sensor and the long-term stability of the sensor due to the high chemical stability of these nanomaterials. The electrical conductivity of the fabricated sensors using metal oxide nanocomposite might also based on the molecular structure and the polymeric medium such as the crystallinity and the long chain polymer [67].
9- Highlight the novelty of this work at the end of the introduction or at the end of the results and discussion
Response: The novelty of this study has been highlighted in the end of introduction as well as the end of results and discussion
Response: The main key finds have been provided in the conclusion section.

Reviewer 2 Report
This article is interesting, the content is clear and to the point.
- In Fig.8a, there are only four data points. To make the results more reliable, there should be no fewer than five data points.
- Authors should have the article reviewed for grammar correctness.
Author Response
Reviewer 2:
Thank you very much for your valuable comments. The manuscript has been carefully revised and the comments have been answered point by point and highlighted by red colour.
This article is interesting, the content is clear and to the point.
- In Fig.8a, there are only four data points. To make the results more reliable, there should be no fewer than five data points.
Response: Figure 8a has been corrected and plotted again using 6 points.
- Authors should have the article reviewed for grammar correctness.
Response: English improvement has been performed throughout the whole text and the corrections highlighted with red colour.

Round 2
Reviewer 1 Report
Authors respond to all comments